# Preclinical Evaluation of Soluble Epoxide Hydrolase Inhibitor AMHDU against Neuropathic Pain

**DOI:** 10.3390/ijms25168841

**Published:** 2024-08-14

**Authors:** Denis Babkov, Natalya Eliseeva, Kristina Adzhienko, Viktoria Bagmetova, Dmitry Danilov, Cynthia B. McReynolds, Christophe Morisseau, Bruce D. Hammock, Vladimir Burmistrov

**Affiliations:** 1Department of Pharmacology & Bioinformatics, Scientific Center for Innovative Drugs, Volgograd State Medical University, Volgograd 400131, Russia; denis.a.babkov@gmail.com (D.B.); vvbagmetova@gmail.com (V.B.); 2Department of Organic Chemistry, Volgograd State Technical University, Volgograd 400005, Russia; danilov.dmitry.vlz@yandex.ru; 3Department of Entomology and Nematology, UC Davis Comprehensive Cancer Center, University of California Davis, Davis, CA 95616, USAchmorisseau@ucdavis.edu (C.M.);

**Keywords:** soluble epoxide hydrolase, inhibitor, urea, adamantane, preclinical evaluation

## Abstract

Inhibition of soluble epoxide hydrolase (sEH) is a promising therapeutic strategy for treating neuropathic pain. These inhibitors effectively reduce diabetic neuropathic pain and inflammation induced by Freund’s adjuvant which makes them a suitable alternative to traditional opioids. This study showcased the notable analgesic effects of compound **AMHDU** (1,1′-(hexane-1,6-diyl)bis(3-((adamantan-1-yl)methyl)urea)) in both inflammatory and diabetic neuropathy models. While lacking anti-inflammatory properties in a paw edema model, **AMHDU** is comparable to celecoxib as an analgesic in 30 mg/kg dose administrated by intraperitoneal injection. In a diabetic tactile allodynia model, **AMHDU** showed a prominent analgesic activity in 10 mg/kg intraperitoneal dose (*p* < 0.05). The effect is comparable to that of gabapentin, but without the risk of dependence due to a different mechanism of action. Low acute oral toxicity (>2000 mg/kg) and a high therapeutic index makes **AMHDU** a promising candidate for further structure optimization and preclinical evaluation.

## 1. Introduction

The impact of neuropathic pain on public health is significant, as it is a common and severe form of chronic pain that affects a considerable portion of the population, particularly middle-aged to older individuals with various underlying causes, especially diabetes mellitus [1,2]. This condition is estimated to be prevalent in 3% to 18% of different populations [3,4]. Neuropathic pain is associated with a higher health burden and is challenging to manage effectively with traditional analgesics like nonsteroidal anti-inflammatory drugs or opioids. Instead, medications like gabapentinoids, tricyclic antidepressants, and serotonin–norepinephrine reuptake inhibitors are recommended as first- and second-line of treatments [4]. However, these treatments in many cases do not provide adequate pain relief for all individuals. Neuropathic pain can lead to a decrease in health-related quality of life, impact daily functioning and contribute to a range of comorbidities, affecting not only physical health but also mental well-being and social aspects. This underscores the importance of understanding the epidemiology of neuropathic pain, its prevalence, associated factors and the need for effective prevention and management strategies to address its impact on public health [5].

Soluble epoxide hydrolase (sEH) plays a crucial role in the metabolism of epoxy–fatty acids (EpFAs), which are natural signaling molecules involved in inflammatory and neuropathic pain pathways [6]. In the context of neuropathic pain, sEH inhibition has been shown to be antinociceptive, meaning it can reduce pain sensation. Studies have demonstrated that inhibiting sEH can attenuate chronic pain in conditions like murine diabetic neuropathy [7]. Inhibition of sEH is associated with enhanced synaptic neurotransmission and plasticity in the prefrontal cortex, contributing to its analgesic effects. Furthermore, sEH inhibitors have been developed as potential therapeutic agents for resolving inflammation and neuropathic pain without the risk of addiction [8]. It has been shown that sEH-produced EpFAs mediate cross-talk between inflammation, oxidative stress and NLRP3 inflammasome, highlighting therapeutic potential of sEH inhibitors in diabetic complications [9]. Understanding the role of sEH and its modulation in pain pathways is crucial for developing novel effective strategies to manage neuropathic pain and improve patient outcomes.

Recent advancements in sEH inhibitors have shown promising developments in both preclinical and clinical studies [10,11,12,13]. These small chemicals have gained attention for their therapeutic potential in various conditions such as cardiovascular diseases, central nervous system disorders and metabolic diseases [14,15]. Notably, sEH inhibitors can help maintain endogenous epoxyeicosatrienoic acid (EETs) levels, offering therapeutic benefits for cardiovascular, central nervous system and metabolic diseases. Some of the compounds that have emerged as sEH inhibitors include *t*-TUCB, which has shown efficacy in promoting the polarization of macrophages to anti-inflammatory M2-type cells and inducing hepatic autophagy [16]. Additionally, PTUPB, a multi-target sEH inhibitor combined with COX-2 inhibition, has demonstrated potential in conditions where eosinophil and pain-associated inflammation coexist. The most advanced compound, EC5026, entered phase Ib clinical trials in 2024, and so far no adverse effects have been reported [10,17,18]. These advancements highlight the diverse applications and potential of sEH inhibitors in addressing a range of health conditions through their anti-inflammatory and analgesic properties.

Previously, we reported 1,1′-(hexane-1,6-diyl)bis(3-((adamantan-1-yl)methyl)urea) (**AMHDU**) as a potent slow tight binding sEH inhibitor (IC_50_ 0.5 nM, *K*_i_ 3.1 nM) [19]. The objective of the present study is to evaluate the anti-inflammatory and analgesic activity of **AMHDU** as a preclinical candidate, including its pharmacokinetic profile and acute oral toxicity.

## 2. Results

### 2.1. Anti-Inflammatory and Analgesic Activity

In the course of the study, it was revealed that a pronounced edema of the paw develops at the sub-plantar injection of Freund’s adjuvant, as evidenced by a reliable increase in its volume (Figure 1). The maximum paw edema (peak of inflammation) developed 4 days after adjuvant administration. The volume of the paw in animals of the control group increased on average 2.3 times and amounted to 0.82 ± 0.07 mL. In the test group, 4 days after adjuvant administration **AMHDU** compound at the dose of 5 mg/kg had no significant effect on the volume of the affected limb, and no statistically significant reduction of the primary exudative reaction under the action of 30 mg/kg **AMHDU** was observed ether, while by comparison, positive control drugs, celecoxib and dexamethasone, yielded greater effects, 51% and 65%, respectively. The experimental results indicate that **AMHDU** had a negligible anti-inflammatory effect.

When analgesic activity was studied (Figure 1), hyperalgesia developed in the group of animals with adjuvant inflammation, and the paw-pulling reflex occurred when the weight reached 308.2 ± 38.4 g, which is almost 2 times less than the initial values. At a dose of 5 mg/kg, **AMHDU** demonstrated no significant suppression of the pain threshold, while at a dose of 30 mg/kg, it increased the paw-pulling threshold by 38% and was comparable to the reference drugs, which failed to provide a significant analgesic effect.

Next, we evaluated the influence of AMHDU in a model of neurogenic pain syndrome. In the control group of intact animals, no signs of development of tactile allodynia were observed, which manifested in the absence of pain response to the application of maximal filament 5.18. In animals with streptozotocin diabetes, initial signs of allodynia were registered on the 5th day, and on the 14th day, the value of 50% threshold of pain response was 4.6 ± 1.3 g in the control group.

The investigated compound at a dose of 10 mg/kg caused a significant increase in pain thresholds, reducing the manifestation of allodynia, while exceeding the indicators of the control group 1.5 times. Statistically significant differences between the compound under study and the comparison drug gabapentin were not revealed (Figure 2).

### 2.2. Pharmacokinetics Nature of the Molecule

**AMHDU** pharmacokinetic was tested i.p. at a 1.25 mg/kg dose in male mice. The dose, formulation and species used are consistent with previous studies using sEH inhibitors so that the PK properties can be compared and evaluated to best select for improved PK properties [20]. The results (Table 1 and Figure 3) indicate that the compound is rapidly absorbed to reach, within an hour, (T_max_) a maximal concentration (C_max_) that is roughly 100 times the **AMHDU** IC_50_ (0.5 nM). The compound is also metabolized and/or eliminated quickly as none is left in the bloodstream 4 h after injection (Figure 3). This is reflected by a half-life (T_half_), which refers to the time required for plasma concentration of a drug to decrease by 50%, of around 10 min. This is lower than for previously described sEH inhibitors [20]. However, the compound blood concentration is well above its IC_50_ for at least 3 h post injection. This is reflected in the compound mean residence time (MRT) of around 1 h (Table 1). The area under the plasma drug concentration–time curve (AUC) reflects the actual body exposure to drug after administration of a dose of the drug. Globally, an AUC of 26 ng·m^−1^·h^−1^ was obtained, indicating a sub-optimal exposure following injection of **AMHDU**. These early screening data using consistent formulation with previous compounds helped select **AMHDU** as the lead compound for development; however, a more complete PK study, especially involving several dosages, routes of administration and species is necessary for further development and usage in vivo of the lead compound. Such a more complete PK study will also benefit from the determination of the effective dose (from animal studies like the one described in Section 2.1) to optimize, through PK/PD, blood concentrations above this threshold for a duration of the disease state.

### 2.3. Acute Oral Toxicity Results

The acute toxicity of the substance by intragastric administration to female outbred rats was studied; the toxicity class, tolerated, toxic, lethal doses and character of toxic effect were determined according to the OECD test No. 423.

As a result of the conducted studies, it was established that the tested substance **AMHDU** at a single intragastric administration to female rats both at a dose of 300 mg/kg and at a limit dose of 2000 mg/kg in the maximum allowable volume of 20 mL/kg does not contribute to death in the early and distant periods of observation. After euthanasia and subsequent necropsy of rats carried out on the 15th day after intragastric administration of the tested substances, no pathological changes in the architectonics of internal organs location were registered. In the study conducted on the 15th day after a single intragastric injection of the tested substances in control animals receiving 50% dimethyl sulfoxide (20 mL/kg), as well as in experimental animals receiving **AMHDU** substance in doses of 300 and 2000 mg/kg, the mass ratios of internal organs did not exceed the normal values (Table 2).

Based on the results of this study, it is concluded that the substance **AMHDU** belongs to class 5 toxicity according to the GHS classification, the LD_50_ of which is in the range >2000–5000 mg/kg.

## 3. Discussion

Extensive studies of soluble epoxide hydrolase inhibitors efficacy on animal models of neuropathic pain have been previously conducted, indicating the high potential for using sEH inhibitors as an alternative to classical analgesics like opioids. sEH inhibitors are potent and efficacious in lowering diabetic neuropathic pain and inflammatory pain. sEH inhibitors are efficacious in lowering the inflammation process, and the pain associated therewith, in a variety of animal species [21]. More specifically, sEH inhibitors lower diabetic neuropathic pain and lipopolysaccharide induced inflammation in animals more potently and efficaciously than celecoxib in rats [22]. sEH inhibitors reduce both inflammation and pain through the stabilization of epoxy–fatty acids, which in turn act through different pathways to affect these two pathologies. The action of EpFAs in regulating inflammation is mostly through reduction of the activation of NF-κB pathway and promotion of resolution [23]. The analgesic effects of sEH inhibition have been reported to involve action on PPAR receptors (α and γ) in a cAMP-dependent manner, as well as reducing cellular ER stress and possibly the binding of some EpFAs to transient receptor potential (TRP) [24].

Herein, we demonstrated a pronounced analgesic effect of compound **AMHDU** in inflammatory and diabetic neuropathy models. Interestingly, **AMHDU** lacks anti-inflammatory activity in a paw edema model but is superior to both dexamethasone and celecoxib as an analgesic agent. Lack of anti-inflammatory activity is not typical for sEH inhibitors and might be attributed to rapid clearance of the **AMHDU** preventing its influence on exudative phase of inflammation. Using the diabetic tactile allodynia model, we observed that **AMHDU** is comparable to standard of care gabapentin in analgesic activity, but unlike the latter lacks addictive potential. Low acute oral toxicity (LD_50_ > 2000 mg/kg in rats) and pharmacologically active dosages (5–30 mg/kg) indicate high therapeutic index of the compound. The pharmacokinetic profile of the compound is suboptimal with T_max_ of 1 h and high clearance rate, which is reflected by low T_half_.

## 4. Materials and Methods

### 4.1. General

All animal procedures were performed in accordance with ethical standards for animal manipulation adopted by the European Convention for the Protection of Vertebrate Animals Used for Experimental and Other Scientific Purposes (1986) and considering the International Recommendations of the European Convention for the Protection of Vertebrate Animals Used for Experimental Research (1997). All sections of this study comply with the ARRIVE Guidelines for Reporting Animal Research [25]. Specific activity and toxicity experiments were performed on 54 non-linear male and 15 female rats weighing 200–230 g, 4.5–5 months old, obtained from the laboratory animal nursery “Rappolovo” LLC of the Russian Academy of Medical Sciences (St. Petersburg, Russia). Male Swiss–Webster mice, 6–8 weeks old, were acquired from Charles River Laboratories (Wilmington, MA, USA). **AMHDU** was synthesized according to Burmistrov et al. (2014) [19].

### 4.2. Anti-Inflammatory and Analgesic Activity Study

Chronic inflammation was modeled by subplantar injection of Freund’s adjuvant (0.1 mL of BCG suspension 2.5 mg/mL in Vaseline oil) into the hind paw of rats. The inflammatory response was assessed by the intensity of inflammatory changes in the joints of the affected limb, expressed in the edema index and by changes in pain sensitivity thresholds. The primary reaction (edema on the injection paw) was evaluated oncometrically on the 4th day after adjuvant injection with plethysmometer (UgoBasile, Gemonio, Italy). The reference drugs dexamethasone (5 mg/kg) and celecoxib (10 mg/kg), as well as the compound **AMHDU** (5 and 30 mg/kg) under study, were dissolved in 10% DMSO and injected intraperitoneally for 4 days, starting from the first day of the experiment 2 times a day. The equivalent volume of solvent was administered to control animals. Animals were assigned to each group randomly (n = 6). The analgesic activity of the substance was assessed using the Randall–Selitto test in the modeling of adjuvant inflammation. Pain threshold was determined against the background of constantly increasing mechanical pressure, which should provide a smooth increase in the load on the inflamed paw before the onset of pain response, using the PAM system (UgoBasile, Italy). The value of the pain threshold was the weight in grams, at the achievement of which the reflex “paw retraction” was manifested.

### 4.3. Tactile Allodynia Evaluation

To evaluate the effect of the compound on the development of neuropathic pain, tactile hyperalgesia was evaluated in diabetic polyneuropathy in rats with 2-week streptozotocin-induced diabetes (35 mg/kg intraperitoneally). Only animals with developed allodynia and hyperalgesia were taken into the experiment, with 50% pain response indices not exceeding 6 g. Groups were formed randomly (n = 6). Tactile allodynia in rats was assessed by registering the pressure at which the animals jerked the right hind paw, avoiding exposure to stimuli of increasing degree. VonFrey hairs, which are 20 monofilaments of nylon filaments of different diameters attached to plastic handles, were applied to the right hind paw in turn from below through the mesh. The comparison drug gabapentin as well as the test compound **AMHDU** dissolved in 10% DMSO was administered intraperitoneally in 10 mg/kg dose 60 min. before the experiment. An equivalent volume of solvent was administered to control animals. The experiment was started with a monofilament labeled 4.31. If there was no response after 5 incidents of touching, the monofilament was successively changed to the next monofilament with higher strength. The test was terminated when the filament labeled 5.18 was reached or 4 trials after the first positive response. The 50% threshold of paw retraction during successive increases and decreases in stimulus strength was determined using the UP-AND-DOWN method.

### 4.4. Pharmacokinetic Study

**AMHDU** was evaluated for plasma exposure in male Swiss–Webster mice (6–8 weeks old from Charles River) with an average weight of 31 ± 0.1 g. Animal experiments were approved by the Animal Use and Care Committee of University of California, Davis. Compounds were formulated in 5% EtOH, 95% PEG400 at a concentration of 0.25 mg/mL. Mice were administered 5 mL/kg for a final concentration of 1.25 mg/kg by intraperitoneal injection. Samples were collected at 0.0825, 0.25, 0.5, 1, 2 and 4 h. For compound detection in plasma, 10 µL whole blood was sampled by tail nick and transferred immediately to 50 µL water containing 0.1% EDTA. Samples were extracted as previously described [26] and stored at −20 °C until analysis. Samples were analysed by LC-MS/MS on an Agilent SL 1200 series LC system (Agilent, Palo Alto, CA, USA) connected to a 4000 Qtrap mass spectrometer (Applied Biosystems Instrument Corporation, Foster City, CA, USA) with Turbo V ion source. Liquid chromatography was performed on a Kinetex R© C18 column (100 × 2.1 mm, 1.7 μm). The mass spectrometer was operated in positive modes with primary MRM scan of 499.377/143.153 and secondary scans of 499.377/149.190 and 499.377/166.204. Peaks were integrated using Analyst software v 1.6.3 (Ab Sciex, Washington, DC, USA) and quantified against a standard curve as previously described [27]. Individual PK parameters were calculated by fitting blood concentrations to a non-compartmental analysis, mixed log-linear AUC model using Kinetica software (Thermo Fisher Scientific, Waltham, MA, USA, version 5.1).

### 4.5. Acute Oral Toxicity Study

The studies were performed in accordance with the rules of Good Laboratory Practice (GLP) [28,29].

Experiments were performed on sexually mature laboratory white mongrel female rats, divided into experimental and control groups. Experimental animals were administered the **AMHDU** substance diluted in 50% dimethyl sulfoxide solution in the recommended GOST 32644-2014 initial dose of 300 mg/kg (n = 6), as well as the maximum dose of 2000 mg/kg (n = 3) in the maximum allowable volume of 20 mL/kg once intra-gastrically. Control animals (n = 6) were injected with dimethyl sulfoxide diluted 1:1 with purified water at volumes identical to those of the experimental animals. The animals were monitored for 2 weeks, continuously for the first 4 h after injection. Routine necropsy of all surviving animals was performed on day 15.

### 4.6. Data Analysis and Statistics

Data analysis and visualization were performed with R 4.3.3 (packages ggplot2 [30], ggpubr [31], ggsignif [32], and patchwork [33]) using RStudio 2023.12.1+402 “Ocean Storm” Release. Data was checked for normality and visualized as median and inter-quartile range unless otherwise noted. Nonparametric Kruskal–Wallis test with Dunn’s post-test was used for multiple comparisons and Mann–Whitney U test for pairwise comparisons.

## 5. Conclusions

Our lead compound AMHDU was shown to effectively reduce pain in vivo. However, its action was limited by less-than-optimal pharmacological properties. Thus, future optimization efforts should be directed towards optimization of stability and retention of the drug to improve its duration of action. Improved analogs will be a subject for full preclinical study, including dose-response and safety.

## Figures and Tables

**Figure 1 ijms-25-08841-f001:**
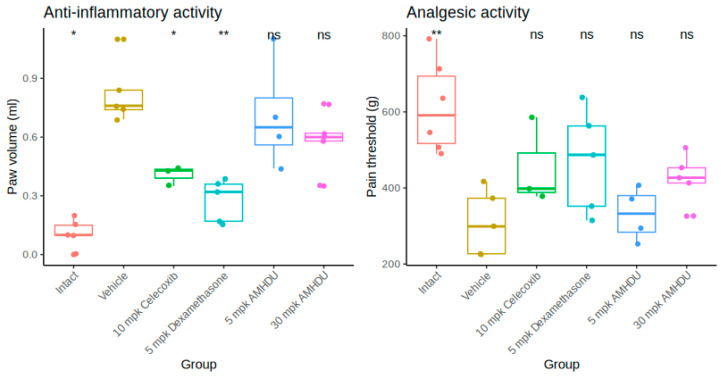
Anti-inflammatory and analgesic activity of **AMHDU** in rat paw edema model (n = 6). Statistical significance is reported vs. vehicle group: * *p* < 0.05, ** *p* < 0.01, ns *p* > 0.05 (Kruskal–Wallis test with Dunn’s post-test).

**Figure 2 ijms-25-08841-f002:**
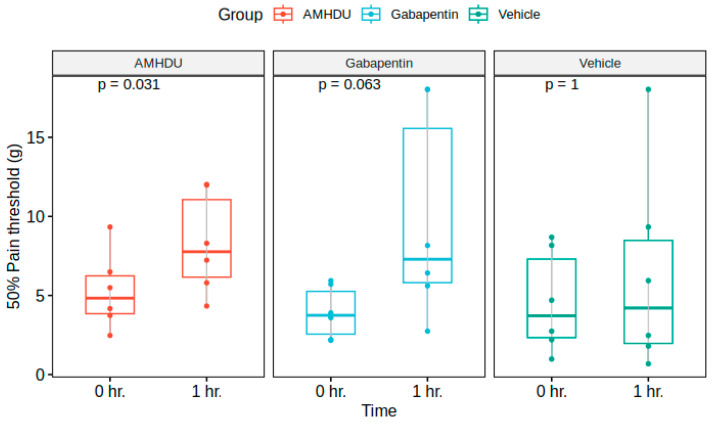
**AMHDU** ameliorates tactile hyperalgesia in STZ-rat neuropathic pain model (n = 6). Statistical significance is reported vs. 0 h. matched values for each group: (Friedman test).

**Figure 3 ijms-25-08841-f003:**
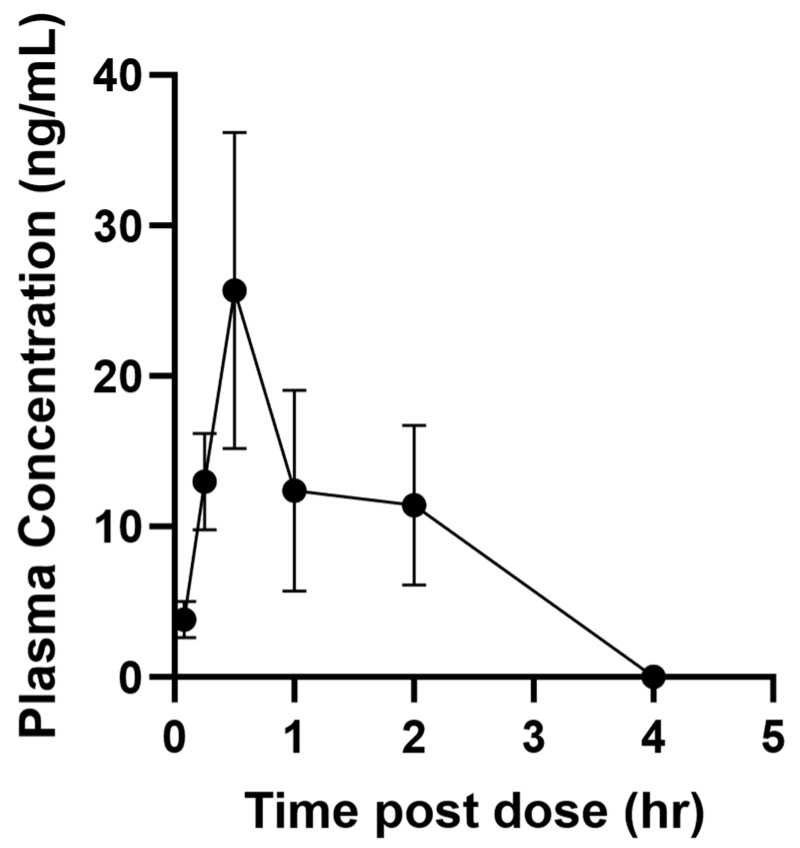
Plasma concentration of mice administered 1 mg/kg of compound **AMHDU** by intraperitoneal injection. Data represent mean plus standard error of n = 4 male mice.

**Table 1 ijms-25-08841-t001:** PK parameters for compound **AMHDU**.

Animal No.	Dose (mg/kg)	C_max_ (ng/mL)	T_max_ (h)	AUC_last_ (ng/mL × h)	T_half_ (h)	MRT (h)
1	1	24.32	2.0	24	0.18	1.51
2	1	53.00	0.5	35	0.16	1.04
3	1	31.93	1.0	31	0.13	1.08
4	1	30.86	0.5	16	0.15	0.55
**m ± SE**		**35.0 ± 12.4**	**1.0**	**26.4 ± 8.6**	**0.16 ± 0.02**	**1.1 ± 0.4**

**Table 2 ijms-25-08841-t002:** Internal organ mass ratios of female rats (%, m ± SD) 2 weeks after a single intragastric administration of **AMHDU** *.

Mass Ratio to Whole Body Weight (%)	Group		
	Vehicle (n = 6)	300 mg/kg AMHDU (n = 6)	2000 mg/kg AMHDU (n = 3)
Brain	0.72 ± 0.02	0.72 ± 0.02 (–0.1)	0.72 ± 0.02 (–0.3)
Heart	0.36 ± 0.01	0.33 ± 0.01 (–6.4)	0.32 ± 0.01 (–10.4)
Lungs	0.63 ± 0.03	0.60 ± 0.04 (–3.5)	0.56 ± 0.03 (–10.7)
Liver	2.96 ± 0.18	3.03 ± 0.12 (2.1)	2.77 ± 0.41 (–6.4)
Kidneys	0.64 ± 0.03	0.62 ± 0.02 (–3.1)	0.64 ± 0.03 (0.4)
Thymus	0.18 ± 0.02	0.17 ± 0.03 (–4.1)	0.16 ± 0.01 (–11.0)
Spleen	0.47 ± 0.05	0.48 ± 0.01 (0.5)	0.43 ± 0.02 (–9.9)
Adrenal glands	0.04 ± 0.003	0.04 ± 0.004 (–8.5)	0.03 ± 0.003 (–14.4)
Ovaries	0.05 ± 0.004	0.05 ± 0.003 (–1.0)	0.06 ± 0.01 (18.3)

* Changes in % compared to the control group are given in parentheses.

## Data Availability

Data are contained within the article.

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
