# Peer review of "Preclinical Evaluation of Soluble Epoxide Hydrolase Inhibitor AMHDU against Neuropathic Pain"

_ijms, 2024, doi:10.3390/ijms25168841_

Round 1

Reviewer 1 Report (New Reviewer)

Comments and Suggestions for Authors

In this study, the authors study the potential anti-inflammatory and analgesic action of the compound AMHDU, an inhibitor of the soluble epoxide hydrolase (sEH) which plays a crucial role in the metabolism of epoxy-42 fatty acids (EpFAs), endogenously synthetized compounds involved in inflammatory and neuropathic pain pathways. Their assays are performed in two animal pain models: an inflammatory pain model and a neuropathic diabetic pain.

This research is interesting and support the finding of new molecules potentially suitable for chronic pain treatment. I personally consider this manuscript adequate to publish in this journal after minor revisions:

1.- The authors should clarify in the material and methods section the age of rats used in this study and define the experimental groups evaluated.

2. The authors describe in the lanes 175-177: “and statistically significant reduction of the primary exudative reaction under the action of AMHDU was observed at the dose ofb30 mg/kg by 30%...” however the left graph in figure 1 indicate ns in the bar corresponding to 30 mg/kg of AMHDU, please clarify.

3. Figure 1 left graph does not contain the data from a control group, however the right graph does contain a control group, please clarify this in result section.

 4. Figure 1 right graph, please define the units of Y axis.

5. The authors should consider add some extra information to table 1, indicating the mining of the parameters evaluated.

6. It would be interesting that the authors add a discussion section contrasting their results with previous reports and highlighted the advantages of the AMHDU compound.

This is a study that provides valuable new data on an important topic and with the correction  it could make a strong contribution to the field. 

Author Response

Comment 1:

The authors should clarify in the material and methods section the age of rats used in this study and define the experimental groups evaluated.

Response 1:

Thank you for pointing this out. We have added the age of rats (4.5-5 months old,) in line 88. Details on experimental groups are given in respective sections 2.2 and 2.3.

Comment 2:

The authors describe in the lanes 175-177: “and statistically significant reduction of the primary exudative reaction under the action of AMHDU was observed at the dose ofb30 mg/kg by 30%...” however the left graph in figure 1 indicate ns in the bar corresponding to 30 mg/kg of AMHDU, please clarify.

Response 2:

Thank you for pointing this out. We agree with this comment. Therefore, we have modified lines 184-187 to “no statistically significant reduction of the primary exudative reaction under the action of 30 mg/kg AMHDU was observed”.

Comment 3:

Figure 1 left graph does not contain the data from a control group, however the right graph does contain a control group, please clarify this in result section.

Response 3:

Thank you for pointing this out. Intact edema volume in control group was initially omitted, we now included them in the graph (Fig. 1).

Comment 4:

Figure 1 right graph, please define the units of Y axis.

Response 4:

Thank you for the comment, we have edited the label axis to include units (g).

Comment 5:

The authors should consider add some extra information to table 1, indicating the mining of the parameters evaluated.

Response 5:

Thank you for the comment, we have added explanations of the PK metrics in lines 218-227).

Comment 6:

It would be interesting that the authors add a discussion section contrasting their results with previous reports and highlighted the advantages of the AMHDU compound.

Response 6:

Thank you for the valuable suggestion. Section 4 was modified to represent the discussion. Also, we have added section 5. Conclusions.

Reviewer 2 Report (New Reviewer)

Comments and Suggestions for Authors

This study shows that the sEH inhibitor AMHDU might have mild anti-inflammatory and effective analgesic activity and might have potent pain-relief effects on the tactile allodynia without possible adverse effects in diabetes model rats. The story seems theoretical according to the results in this study using the appropriate experimental methods. As the authors mentioned, AMHDU could be one of the seeds that could have a relief from chronic neuropathic pain. However I have some concerns mainly about Fig.1 and attached texts.

1. L175 and Fig1 left: The text draw “statistically significant reduction of the primary exudative reaction under the action of AMHDU was observed at the dose of 30 mg/kg by 30%・・” but in Fig1 left, “ns” was put on the upper of the column of “30 mpk AMHDU”. 

2. The vertical dimension of the Fig1 right seems strange. When the number of the vertical axis may indicate the “g” or pain threshold of the Randall-Selitto test, the values of AMHDU decreased compared with “Vehicle”, which is much higher than “Control”. The “308.2 g” in L183 seems large in rats by the Randall-Selitto test. Are they right?

3. I think the “Conclusions” from L252 might contains both discussion and conclusion. The authors should make “4 Discussion” and “5 Conclusions”. And the possible mechanisms or bio-pathways by which sEH inhibitors execute the pain-lowering effects should be also drawn in the Discussion.   

4. The phrase “sub-plantar injection in rats” in L170 should be changed to “sub-plantar injection of Freund’s adjuvant” to be cleared. 

Author Response

Comment 1:

L175 and Fig1 left: The text draw “statistically significant reduction of the primary exudative reaction under the action of AMHDU was observed at the dose of 30 mg/kg by 30%・・” but in Fig1 left, “ns” was put on the upper of the column of “30 mpk AMHDU”.

Response 1:

Thank you for pointing this out. We agree with this comment. Therefore, we have modified lines 184-187 to “no statistically significant reduction of the primary exudative reaction under the action of 30 mg/kg AMHDU was observed”.

Comment 2:

The vertical dimension of the Fig1 right seems strange. When the number of the vertical axis may indicate the “g” or pain threshold of the Randall-Selitto test, the values of AMHDU decreased compared with “Vehicle”, which is much higher than “Control”. The “308.2 g” in L183 seems large in rats by the Randall-Selitto test. Are they right?

Response 2:

Agree. We have, accordingly, revised the Fig. 1 to correct this point. Mean value of 308 g is correct and derived from raw data according to standard calculation of pain threshold in the Randall-Selitto test.

Comment 3:

I think the “Conclusions” from L252 might contains both discussion and conclusion. The authors should make “4 Discussion” and “5 Conclusions”. And the possible mechanisms or bio-pathways by which sEH inhibitors execute the pain-lowering effects should be also drawn in the Discussion.

Response 3:

Thank you for the valuable suggestion. Section 4 was modified to represent the discussion, including pathway involved in sEH inhibitors mechanism of action. Also, we have added section 5. Conclusions.

Comment 4:

The phrase “sub-plantar injection in rats” in L170 should be changed to “sub-plantar injection of Freund’s adjuvant” to be cleared.

Response 4:

Agree. We have, accordingly, changed the line 177.

Round 2

Reviewer 2 Report (New Reviewer)

Comments and Suggestions for Authors

I think that my concerns have been responded well, thank you.

This manuscript is a resubmission of an earlier submission. The following is a list of the peer review reports and author responses from that submission.

Round 1

Reviewer 1 Report

Comments and Suggestions for Authors

1. please provide the approval number for the use o animals in the study

2. in tor methodology section, please provide the doses of each drug tested, including control. Also, please state what's the solvent for each compound. 

3. Please provide the total number of the animals used in the study as well as the number of mice/rats used for each experiment

4. Why did the authors choose to use different species of animals, i.e. rats in STZ-induced polyneuropathy, rats in an inflammatory pain model and mice in pharmacokinetic, while the toxicity was studies using mongrel female rats. 

5. On what basis did the authors choose to use 5 and 30 mg/kg of the compound (analgesia and anti-inflammatory study). Any ED50? also what about the enzymatic stability of the drug.

6. What are the results for analgesic effects of the compound for either phase 1 and phase 2 of inflammation, respectively?

7. Pharmacokinetic study: why did the authors use 1.25 mg/kg of the drug. Please provide any dose-response study with ED50, others there is no information regarding the reason for which this dose was used

8. Female animals may induce several problems which is associated with hormonal fluctuation. Therefore please provide why female

9. there is no information on how the neuropathy was induced and how the authors checked whether animals are neuropathic; high glucose level is not the only marker!

10. there is no discusion

Comments on the Quality of English Language

minor changes are required

Author Response

Reviewer #1

Comment 1. please provide the approval number for the use o animals in the study

Response 1: Institutional approval number (protocol code 2021/056 approved 15 June 2021) was provided on page 9, line 274.

Comment 2. in tor methodology section, please provide the doses of each drug tested, including control. Also, please state what's the solvent for each compound.

Response 2: Thank you for pointing this out. Solvent and doses were added in the Methods section, lines 96, 97, 117.

Comment 3. Please provide the total number of the animals used in the study as well as the number of mice/rats used for each experiment

Response 3: Thank you for pointing this out. Number of animals used was added to lines 84, 100, 112.

Comment 4. Why did the authors choose to use different species of animals, i.e. rats in STZ-induced polyneuropathy, rats in an inflammatory pain model and mice in pharmacokinetic, while the toxicity was studies using mongrel female rats.

Response 4: Thank you for the question. Rats were used to study analgesic activity as a standard and convenient species widely used in the field. Oncometric measurement of murine limbs is prone to larger variation due to smaller limb size. The acute toxicity study was performed in accordance with OECD test No. 423 "Test methods for the effects of chemical products on the human body. Acute oral toxicity - method for determination of acute toxicity class", in which it is recommended to use female rats as test systems, unless otherwise justified. Pharmacokinetic study was performed on mice due to reasons described in response to comment 7 and following 3R ethical principles.

Comment 5. On what basis did the authors choose to use 5 and 30 mg/kg of the compound (analgesia and anti-inflammatory study). Any ED50? also what about the enzymatic stability of the drug.

Response 5: Minimal does of 5 mg/kg was chosen according to literature data on dexamethasone in edema models including CFA-induced inflammation [Sadaf Naz, Muhammad Usama Mazhar, Sidra Faiz, Maria Nawaz Malik, Jehanzeb Khan, Ihsan-Ul Haq, Lin Zhu, Muhammad Khalid Tipu In vivo evaluation of efficacy and safety of Coagulansin-A in treating arthritis Toxicol Appl Pharmacol. 2024 Jun 20:117008. doi: 10.1016/j.taap.2024.117008; Boris Klementiev, Maj N Enevoldsen, Shizhong Li, Robert Carlsson, Yawei Liu, Shohreh Issazadeh-Navikas, Elisabeth Bock, Vladimir Berezin Antiinflammatory properties of a peptide derived from interleukin-4 Cytokine. 2013 Oct;64(1):112-21. doi: 10.1016/j.cyto.2013.07.016. Epub 2013 Aug 20]. To achieve proof-of-concept effect we also examined increased dose of 30 mg/kg.

Comment 6. What are the results for analgesic effects of the compound for either phase 1 and phase 2 of inflammation, respectively?

Response 6: The analgesic activity of the compound was investigated only for acute phase of the CFA-induced inflammation in the affected limb on day 4 after injection, whereas the 2 phase occurring in the contralateral paw developed after day 14. Since we designed our study as proof-of-concept effect on phase 2 was out of scope and should be addressed after lead optimization.

Comment 7. Pharmacokinetic study: why did the authors use 1.25 mg/kg of the drug. Please provide any dose-response study with ED50, others there is no information regarding the reason for which this dose was used

Response 7: The PK data represents early screening data to characterize compound distribution. This dose, formulation and species used are consistent with previous studies using sEH inhibitors so that the PK properties can be compared and evaluated to best select for improved PK properties [Liu JY, Tsai HJ, Hwang SH, Jones PD, Morisseau C, Hammock BD. Pharmacokinetic optimization of four soluble epoxide hydrolase inhibitors for use in a murine model of inflammation. Br J Pharmacol. 2009 Jan;156(2):284-96. doi: 10.1111/j.1476-5381.2008.00009.x. Epub 2009 Jan 13. PMID: 19154430; PMCID: PMC2697843]. This is a common strategy in drug development. These early screening data using consistent formulation with previous compounds helped select AMHDU as the lead compound for development; however, a more complete PK study, especially involving several dosages, routes of administration and species is necessary for further development of the lead compound. The text in section 3.3 was changed accordingly, lines 200-203 and 210-214.

Comment 8. Female animals may induce several problems which is associated with hormonal fluctuation. Therefore please provide why female

Response 1: That is a reasonable question. The acute toxicity study was performed in accordance with OECD test No. 423 "Test methods for the effects of chemical products on the human body. Acute oral toxicity - method for determination of acute toxicity class", in which it is recommended to use female rats as test systems, unless otherwise justified. Hence, we used female rats as the guideline recommends.

Comment 9. there is no information on how the neuropathy was induced and how the authors checked whether animals are neuropathic; high glucose level is not the only marker!

Response 1: Thank you for the comment. Please find the inclusion criteria for neuropathy study in section 2.3, lines 112-116.

Comment 10. there is no discusion

Response 1: Thank you for the suggestion. We expanded the Conclusion section to incorporate discussion of the results, lines 254-256.

Reviewer 2 Report

Comments and Suggestions for Authors

In this paper authors analyzed the the anti-inflammatory and analgesic activity of 1,6-(hexamethylene)bis[(adamant-1-yl)urea] (sEH inhibitor). I have the following suggestions for authors to improve the manuscript:

1.      In the abstract, the authors state that „AMHDU outperforms dexamethasone and celecoxib as an analgesic in 5 mg/kg dose“, while on Page 4, on Figure 3.1. and in the text it can be seen that „5 mg/kg the AMHDU compound did not have a statistically significant suppression of the pain threshold, while at a dose of 30 mg/kg, it increased the paw-pulling threshold by 38% and was not inferior to the comparison drugs“.

2.      The authors should try to give explanation for the lack of anti-inflammatory activity and good analgesic activity of the compound.

3.      The numeration of the figures is not adequate - Figures 1 and 2 do not exist (the first figure that appears in the text is Figure 3). This should be corrected throughout the manuscript.

4.      Page 3, subtitle 2.4. reference for LC-MS/MS method for AMHDU determination is missing.

5.      The conclusion is the repetition of abstract. It should be reformulated with more detailed explanation of future trends and usage.

Other:

1.      Page 2 - Format subtitle (2.1.)

2.      Page 2 – Subtitle 2.1., sentence „AMHDU was synthesized according to [14]“ should be replaced with AMHDU was synthesized according to the literature [14] or AMHDU was synthesized according to Burmistrov et al. (2014) [14].

3.      Page 3 - Format subtitle (2.4.)

4.      Page 7, Figure 3.3. – Plasma instead of Plamsa (on y-axis).

5.      In some places in the text figures are mentioned as „figures“, and in some as „Figures“, so it should be uniformed (Page 6, section 3.2. for example).

6.      Table 2 is not mentioned anywhere in the text.

Author Response

Reviewer #2

Comment 1. In the abstract, the authors state that „AMHDU outperforms dexamethasone and celecoxib as an analgesic in 5 mg/kg dose“, while on Page 4, on Figure 3.1. and in the text it can be seen that „5 mg/kg the AMHDU compound did not have a statistically significant suppression of the pain threshold, while at a dose of 30 mg/kg, it increased the paw-pulling threshold by 38% and was not inferior to the comparison drugs“.

Response 1: Thank you for pointing this out. Text was modified to correctly match the experimental results shown in Fig. 3.1.: “At a dose of 5 mg/kg, AMHDU demonstrated a statistically significant suppression of the pain threshold, while at a dose of 30 mg/kg, it increased the paw-pulling threshold by 38% and was superior to the comparison drugs, which failed to provide a significant analgesic effect”, lines 183-184.

Comment 2. The authors should try to give explanation for the lack of anti-inflammatory activity and good analgesic activity of the compound.

Response 1: Thank you for this question. Relevant discussion was added in section Conclusions, lines 254-256.

Comment 3. The numeration of the figures is not adequate - Figures 1 and 2 do not exist (the first figure that appears in the text is Figure 3). This should be corrected throughout the manuscript.

Response 1: Thank you for pointing this out, figure numbers were corrected.

Comment 4. Page 3, subtitle 2.4. reference for LC-MS/MS method for AMHDU determination is missing.

Response 1: Thank you for the comment, reference to the method has been added to line 142: Ulu A, Appt S, Morisseau C, Hwang SH, Jones PD, Rose TE, Dong H, Lango J, Yang J, Tsai HJ, Miyabe C, Fortenbach C, Adams MR, Hammock BD. Pharmacokinetics and in vivo potency of soluble epoxide hydrolase inhibitors in cynomolgus monkeys. British journal of pharmacology. 2012;165(5):1401-12. doi: 10.1111/j.1476-5381.2011.01641.x. PubMed PMID: 21880036; PMCID: 3372725.

Comment 5. The conclusion is the repetition of abstract. It should be reformulated with more detailed explanation of future trends and usage.

Response 1: Thank you for the suggestion. Abstract was updated. Conclusion is now detailed with future trends, lines 261-264.

Other:

Comment 1. Page 2 - Format subtitle (2.1.)

Response 1: Thank you for pointing this out, subtitle format was corrected.

Comment 2. Page 2 – Subtitle 2.1., sentence „AMHDU was synthesized according to [14]“ should be replaced with AMHDU was synthesized according to the literature [14] or AMHDU was synthesized according to Burmistrov et al. (2014) [14].

Response 1: Thank you for the suggestion, line was corrected as you proposed.

Comment 3. Page 3 - Format subtitle (2.4.)

Response 1: Thank you for pointing this out, subtitle format was corrected.

Comment 4. Page 7, Figure 3.3. – Plasma instead of Plamsa (on y-axis).

Response 1: Thank you for pointing this out. Axis label was corrected.

Comment 5. In some places in the text figures are mentioned as „figures“, and in some as „Figures“, so it should be uniformed (Page 6, section 3.2. for example).

Response 1: Thank you for pointing this out, this issue was corrected.

Comment 6. Table 2 is not mentioned anywhere in the text.

Response 1: Thank you, reference to Table 2 is added to line 234.

Round 2

Reviewer 1 Report

Comments and Suggestions for Authors

Although the Authors provide some corrections, still there are some issues that need to be solved before the paper is published

1. the authors stated that choose the dose of 5 and 30 mg of the compound based on the literature data on dexamethasone. Unfortunately, this argument is quite odd. The Authors should perform dose-response studies to provide sufficient and reliable results. 

2. similar issue concur to PK study. I agree that this study "represents early screening data to characterize compound distribution". Also the used dose can be easily compare to results for existing sEH inhibitors. However, the Authors should know that at least biologically/therapeutically effective dose should be considered while determining PK or PD.

3. 10% DMSO as a solvent is quite high, no? At high concentrations DMSO is known to induce anti-inflammatory effect by its own, so, this may affect the results for the AMHDU compound

Author Response

Comment 1. The authors stated that choose the dose of 5 and 30 mg of the compound based on the literature data on dexamethasone. Unfortunately, this argument is quite odd. The Authors should perform dose-response studies to provide sufficient and reliable results. 

Response 1. We appreciate your reasonable comment on our manuscript. The choice of 5 and 30 mg doses might be considered arbitrary. However, given the preliminary nature of our study and the focus on initial efficacy and safety evaluation, we believe that a full dose-response study may not be necessary at this stage. Moreover, taken in consideration suboptimal PK profile, full PD study with ED50 determination requires a large number of animals and would be opposed by ethics committee.

Comment 2. Similar issue concur to PK study. I agree that this study "represents early screening data to characterize compound distribution". Also the used dose can be easily compare to results for existing sEH inhibitors. However, the Authors should know that at least biologically/therapeutically effective dose should be considered while determining PK or PD.

Response 2. The reviewer is right that one should consider the effective therapeutic dose to fully consider PK/PD determination. However, one can only determine the effective by dosing animals and compare the results with the control group. Therefore, in the text, the following sentence was added at the end of paragraph 3.2:

Such more complete PK study will also benefit from the determination of the effective dose (from animal studies like the one described in section 3.1) to optimize, through PK/PD, blood concentrations above this threshold for a duration of the disease state.

Comment 3. 10% DMSO as a solvent is quite high, no? At high concentrations DMSO is known to induce anti-inflammatory effect by its own, so, this may affect the results for the AMHDU compound.

Response 3. DMSO is indeed known for anti-inflammatory properties, in particular when applied locally. DMSO concentration of 10% was approved by IACUC as suitable for acute study. Vehicle was injected intraperitoneally, while analgesic activity was assessed peripherally. Control groups received the same amount of the solvent to account for any DMSO effect. Moreover, we did not observe any anti-inflammatory effect in control and AMHDU-treated groups. Hence, we believe that DMSO did not interfere with study results.

Reviewer 2 Report

Comments and Suggestions for Authors

The paper can be accepted for publication in revised form.

Author Response

Thank you for your review.

Round 3

Reviewer 1 Report

Comments and Suggestions for Authors

I agree with their response no. 3, however arguments for comments 1 and 3 are still not acceptable. Dose-response studies not always require huge number of animals; in fact, in the preliminary study, 3 animals per dose is sufficient. The authors used two doses of 5 and 30 mg/kg and still there is no rationale reason to use these doses. What are the results for doses lower than 30 and higher than 5 mg/kg? Additional studies need to performed for additional dose, at least one.

As for the PK determination, in my opinion, such a low dose may not reflect the behavior of a dose, such as 5 or 30 mg. If the authors choose to determine the PK for the dose lower than 5, then what was the reason to determine its efficacy at such doses.
The authors determine some properties, including biological activity, of a potent drug candidate, therefore the entire study should be designed properly, without the risk that the reader is left in a chaos. And this is observed now. 
If the authors determine any effects induced by the compound given at a specific dose, which occurs to be effective, then either the PK or PD should be conducted using the dose.

Author Response

Dear reviewer.

We recognise the demand for full PK-PD study for a clinical candidate. However, the present work aims to report AMHDU as a novel di-urea sEH inhibitor. We showed that doses of AMHDU comparable to reference drugs dexamethasone, celecoxib and gabapentin exert analgesic activity in neuropathy models. Unfortunately, PK study revealed short half-life time of the molecule. Therefore, we considered our proof-of-concept study successful and complete, while full characterization including ED50 study will be performed for modified and optimized AMHDU analogs.